Effect of vascular resection for perihilar cholangiocarcinoma: a systematic review and meta-analysis

Liu Yong 1
Li Guangbing 2
Lu Ziwen 1
Wang Tao 1
Yang Yang 1
Wang Xiaoyu 1
Liu Jun 1 2 dr_liujun1967@126.com
1 Department of Liver Transplantation and Hepatobiliary Surgery, Shandong Provincial Hospital, Cheeloo College of Medicine, Shandong University , Jinan, Shandong Province , China
2 Department of Liver Transplantation and Hepatobiliary Surgery, Shandong Provincial Hospital Affiliated to Shandong First Medical University , Jinan, Shandong Province , China
Syn Nicholas
Electronic publication date: 2021 Sep 23
Publication date: 2021
Volume: 9
Electronic Location ID: e12184
Received 2021 Apr 29; Accepted 2021 Aug 29
Copyright: © 2021 Liu et al.
Copyright year: 2021
Copyright holder: Liu et al.
License: This is an open access article distributed under the terms of the Creative Commons Attribution License, which permits unrestricted use, distribution, reproduction and adaptation in any medium and for any purpose provided that it is properly attributed. For attribution, the original author(s), title, publication source (PeerJ) and either DOI or URL of the article must be cited.
License URL: https://creativecommons.org/licenses/by/4.0/

Keywords: Perihilar cholangiocarcinoma, Vascular resection, Portal vein resection, Hepatic artery resection, Mortality, Morbidity, Long-term survival

Funding: National Natural Science Foundation of China 81373172 and 81770646 This study was supported by the National Natural Science Foundation of China (Nos. 81373172, 81770646). The funders had no role in study design, data collection and analysis, decision to publish, or preparation of the manuscript.

==============================
Objective

To evaluate the effect of vascular resection (VR), including portal vein resection (PVR) and hepatic artery resection (HAR), on short- and long-term outcomes in patients with perihilar cholangiocarcinoma (PHC).

Background

Resection surgery and transplantation are the main treatment methods for PHC that provide a chance of long-term survival. However, the efficacy and safety of VR, including PVR and HAR, for treating PHC remain controversial.

Methods

This study was registered at the International Prospective Register of Systematic Reviews (CRD42020223330). The EMBASE, PubMed, and Cochrane Library databases were used to search for eligible studies published through November 28, 2020. Studies comparing short- and long-term outcomes between patients who underwent hepatectomy with or without PVR and/or HAR were included. Random- and fixed-effects models were applied to assess the outcomes, including morbidity, mortality, and R0 resection rate, as well as the impact of PVR and HAR on long-term survival.

Results

Twenty-two studies including 4,091 patients were deemed eligible and included in this study. The meta-analysis showed that PVR did not increase the postoperative morbidity rate (odds ratio (OR): 1.03, 95% confidenceinterval (CI): [0.74–1.42], P = 0.88) and slightly increased the postoperative mortality rate (OR: 1.61, 95% CI [1.02–2.54], P = 0.04). HAR did not increase the postoperative morbidity rate (OR: 1.32, 95% CI [0.83–2.11], P = 0.24) and significantly increased the postoperative mortality rate (OR: 4.20, 95% CI [1.88–9.39], P = 0.0005). Neither PVR nor HAR improved the R0 resection rate (OR: 0.70, 95% CI [0.47–1.03], P = 0.07; OR: 0.77, 95% CI [0.37–1.61], P = 0.49, respectively) or long-term survival (OR: 0.52, 95% CI [0.35–0.76], P = 0.0008; OR: 0.43, 95% CI [0.32–0.57], P < 0.00001, respectively).

Conclusions

PVR is relatively safe and might benefit certain patients with advanced PHC in terms of long-term survival, but it is not routinely recommended. HAR results in a higher mortality rate and lower overall survival rate, with no proven benefit.

Introduction

Cholangiocarcinoma is a rare adenocarcinoma that originates from the epithelial cells of bile ducts. Perihilar cholangiocarcinoma (PHC) is the main type of cholangiocarcinoma, accounting for 50% to 67% of cases (Nakeeb et al., 1996; DeOliveira et al., 2007; Alvaro et al., 2020). The prognosis of PHC is generally poor because of its anatomical location and aggressive biology. Resection surgery and transplantation are the main treatment methods for PHC that provide a chance of long-term survival (Ebata et al., 2018). The median overall survival (OS) of patients with PHC who undergo curative resection varies from 19 to 39 months (Popescu & Dumitrascu, 2014).

The objective of surgery is to achieve R0 resection. However, PHC usually adheres to or is surrounded by vessels, such as the portal vein or hepatic artery, which makes curative resection difficult to achieve. Therefore, to achieve R0 resection, vascular resection (VR) can be performed during the operation. It has been reported that the proportion of VR during PHC surgery ranges from 15% to 38% (Higuchi et al., 2019; Mizuno et al., 2020; Schimizzi et al., 2018; She et al., 2020; Yu et al., 2017). VR refers to portal vein resection (PVR), hepatic artery resection (HAR) or both. Although VR is performed in many circumstances, controversy still exists. For PVR, portal vein involvement by PHC was previously considered a sign of unresectability (Ramos, 2013). With the development of surgical techniques, PVR has been performed at several clinical centers (Ebata et al., 2003; Miyazaki et al., 2007; Hemming et al., 2011). However, the efficacy and safety of PVR for PHC are controversial. Ebata et al. (2003) reported that combined portal vein and liver resection can offer long-term survival to some selected patients with advanced PHC. However, Hoffmann et al. (2015) found that PVR greatly increased the perioperative morbidity rate and had no benefit for PHC in terms of the oncologic outcomes. In addition, surgical resection with simultaneous HAR for PHC is a demanding procedure (Miyazaki et al., 2007; Peng et al., 2016; Shimada et al., 2003; Yamanaka et al., 2001). Similar to PVR, attitudes toward HAR remain inconsistent. Miyazaki et al. (2007) reported that HAR had no beneficial effect on prognosis and led to an increase in the perioperative morbidity and mortality rates; thus, the use of HAR may not be justified. Nagino et al. (2010) demonstrated that major hepatectomy with HAR could offer a better chance of long-term survival in selected PHC patients.

To date, several meta-analyses have been performed to evaluate the efficacy and safety of VR for PHC patients; however, the results of these studies were inconsistent. By including 2,457 patients, Abbas & Sandroussi (2013) found that PVR may result in survival benefits for some patients with advanced PHC, which was similar to Chen’s study (Chen, Ke & Chen, 2014). However, Wu et al. (2013) and Yu et al. (2014) found that PVR increases postoperative mortality and morbidity and worsens long-term survival; thus, surgical decisions should be made cautiously. For HAR, Abbas & Sandroussi (2013) and Yu et al. (2014) found that HAR is associated with increased mortality and morbidity without proven survival benefits for PHC patients. In a recent guideline for cholangiocarcinoma from Italy (Cholangiocarcinoma Working Group, 2020), PVR was recommended when there was unilateral portal vein invasion. However, the recommendation for PVR in this study was limited with a low quality of evidence due to the small number of related studies. Further, hardly any attention was given to HAR in the Italian study. Given these conflicting recommendations, the efficacy and safety of PVR and HAR for treating PHC patients need to be further clarified.

The aim of this study was to systematically review and statistically evaluate the effect of VR, including PVR and HAR, on short- and long-term outcomes in PHC patients.

Materials and Methods

Search strategy

This meta-analysis was performed in accordance with the guidelines and review protocols of the Preferred Reporting Items for Systematic Reviews and Meta-analysis (PRISMA) statement (Liberati et al., 2009). This study was registered at the International Prospective Register of Systematic Reviews (CRD42020223330). Two authors (Y.L. and G.B.L.) conducted a literature search independently using the EMBASE, PubMed, and Cochrane Library databases up to November 28, 2020. The search terms were “hilar cholangiocarcinoma”, “Klatskin’s tumour”, “hepatectomy”, “hepatic artery”, “portal vein” and “vascular resection”. Two authors (Y.L. and G.B.L.) independently reviewed the titles, abstracts and full texts for eligibility on the basis of predesigned inclusion and exclusion criteria. Disagreements were settled through consensus or by the judgment of a third author (Z.W.L). A description of the search strategy is shown in our evidence report (Table S2). To avoid omission of other studies that were not indexed, the reference lists of the included studies were also reviewed.

Eligibility criteria

The inclusion criteria were as follows: (1) humans were used as the research objects; (2) full-text articles published in English; (3) all included subjects were diagnosed with PHC; and (4) all enrolled patients underwent curative surgery, with or without resection of the portal vein or hepatic artery. Records were excluded if they were classified as a case report or letter or if the full text was not available. Studies with inadequate data were excluded. Studies including other malignancies, such as gallbladder cancer, hepatic carcinoma or distal cholangiocarcinoma, were also excluded. In the case of duplicate studies, the latest or most integrated data were chosen for analysis.

Data extraction

Two independent reviewers extracted the following attainable data from the included studies: first author, country, year of publication, inclusive period of study, number of patients, Bismuth-Corlette stage, intraoperative blood loss, 90-day mortality, total morbidity, staging of Union for International Cancer Control Unites (UICC), vascular invasion rate, lymph node metastasis rate, median survival time, 1-, 3-, and 5-year OS, 1-, 3-, and 5-year disease-free survival (DFS), and hazard ratios (HRs) with 95% confidence intervals (CIs) for OS. HRs were obtained in two ways: (1) acquired directly from the article or (2) obtained from Kaplan-Meier survival curves following the methods reported by Tierney et al. (2007) and using the Engauge Digitizer version 4.1 (SourceForge, Boston, MA, USA).

Quality assessment

The study quality was assessed using the 9-score system of the Newcastle–Ottawa Scale (NOS) (Stang, 2010). The assessment was based on three aspects: (I) selection; (II) comparability; and (III) outcome. A follow-up duration of at least 2 years was considered adequate. The score provides an assessment of bias for the included studies.

Statistical analysis

The primary purpose of this study was to evaluate the effect of PVR and HAR on long-term outcomes in PHC patients, and the statistical indicators included 1-, 3-, and 5-year OS and 1-, 3-, and 5-year DFS. The secondary purpose of this study was to evaluate the safety of PVR and HAR for PHC patients, and the statistical indicators included 90-day mortality, overall morbidity and the posthepatectomy liver insufficiency (PHI) rate. The 90-day mortality rate included the number of patients who died within 90 days after surgery but excluded the number of patients who died during the operation. Overall morbidity was recorded according to the types of postoperative complications, including intra-abdominal abscess, PHI, bile leakage, vascular complications, etc. (Dindo, Demartines & Clavien, 2004). Since there was no uniform definition of PHI in the included studies, the PHI rate could only be determined based on individual study reports.

Dichotomous categorical variables were analyzed using the Mantel–Haenszel test. Continuous categorical variables were analyzed using the inverse variance test. The results were expressed using forest plots and presented as odds ratios (ORs) and mean differences (MDs) and 95% CIs. Heterogeneity among studies was assessed using the Cochrane Q-test and P-value. Statistically significant heterogeneity was defined as I 2 > 50% or a chi-squared P-value < 0.1. When heterogeneity was significant, a random-effects model was applied; otherwise, a fixed-effects model was used. A “leave-one-out” sensitivity analysis was conducted to identify the source of heterogeneity when significant heterogeneity was present. Funnel plots were used to evaluate the presence of significant publication bias.

The data syntheses in this meta-analysis were performed using RevMan 5.4 and R software (version 4.0.3). A two-sided P < 0.05 was deemed to indicate statistical significance.

Results

Literature search

As shown in Fig. 1, 1,693 records were incipiently included in our search. After the removal of duplicate publications, 1,174 studies remained for title and abstract screening, and 642 records and 422 records were excluded based on title reading and abstract screening, respectively. Subsequently, 110 full texts were assessed for eligibility. Among them, 88 records were further excluded for the following reasons: not in English (n = 7); abstract form only (n = 38); contained other malignancies or benign tumors (n = 2); reconstruction or no reconstruction as comparison item (n = 2); inadequate data (n = 7); and case reports (n = 32). Finally, 22 studies (Higuchi et al., 2019; Mizuno et al., 2020; Schimizzi et al., 2018; She et al., 2020; Yu et al., 2017; Peng et al., 2016; Miyazaki et al., 2007; Nagino et al., 2010; De Jong et al., 2012; Dumitraşcu et al., 2017; Ebata et al., 2003; Hemming et al., 2011; Hoffmann et al., 2015; Igami et al., 2010; Klempnauer et al., 1997; Kondo et al., 2004; Lee et al., 2010; Matsuyama et al., 2016; Muñoz et al., 2002; Song et al., 2009; Tamoto et al., 2014; Wang et al., 2015) including 4091 PHC patients were eligible for this systematic review and meta-analysis.

Figure 1 Flow chart showing the study selection process.

Study characteristics

Study level characteristics are shown in Table 1. All studies were cohort studies published between 1997 and 2020. The total number of patients enrolled was 4,091, and the sample capacities of these studies ranged from 28 to 787 patients. In this meta-analysis, the rates of PVR during curative surgery for PHC varied from 11% to 73%, with an average rate of 27% (Higuchi et al., 2019; Mizuno et al., 2020; Schimizzi et al., 2018; She et al., 2020; Yu et al., 2017; Peng et al., 2016; Miyazaki et al., 2007; Nagino et al., 2010; De Jong et al., 2012; Dumitraşcu et al., 2017; Ebata et al., 2003; Hemming et al., 2011; Igami et al., 2010; Klempnauer et al., 1997; Kondo et al., 2004; Lee et al., 2010; Matsuyama et al., 2016; Muñoz et al., 2002; Song et al., 2009; Tamoto et al., 2014; Wang et al., 2015). Compared to PVR, HAR was relatively rare and performed in only 10% of all enrolled patients.

Table 1 Studies included in the current meta-analysis.

Study (Year)	Country	Period	No of Patients	Male, %	Age (median or mean)	Blood loss
(ml)	90-day mortality, %	Overall morbidity, % (%)	R0, %	Hepatic insufficiency, %	Lymph node metastasis, %	Vascular invasion, %	III, IV of
UICC, % staging, %	5-year survival, %	
Wang et al. (2015)	China	2005–2012	PVR:16	4 (25%)	53	980 ± 511	0 (0%)	6 (38%)	NR	0 (0%)	NR	NR	NR	25%	
			HAR:24	18 (75%)	60	1,175 ± 713	1 (4%)	10 (42%)	NR	1 (6%)	NR	NR	NR	25%	
			No VR:114	70 (61%)	57	527 ± 596	4 (4%)	40 (35%)	NR	2 (2%)	NR	NR	NR	36%	
Dumitraşcu et al. (2017)	Romania	1996–2014	PVR:21	17 (81%)	56	3,475 ± 2,925	2 (10%)	NR	15 (71%)	4 (19%)	12 (57%)	NR	NR	26%	
			No VR:102	53 (52%)	59	400 ± 2,483	5 (5%)	NR	80 (78%)	27 (26%)	36 (35%)	NR	NR	28%	
Ebata et al. (2003)	Japan	1979–2000	PVR:52	35 (67%)	60.3	NR	NR	44 (85%)	36 (69%)	14 (27%)	29 (56%)	25 (48%)	52 (100%)	10%	
			No VR:108	81 (75%)	60.2	NR	NR	85 (79%)	95 (88%)	21 (19%)	43 (40%)	47 (44%)	20 (19%)	37%	
Nagino et al. (2010)	Japan	1997–2009	PVR:92	NR	60	NR	NR	NR	NR	NR	NR	NR	NR	30%	
			HAR:62	NR	60	NR	NR	NR	NR	NR	NR	NR	NR	30%	
			No VR:211	NR	NR	NR	NR	NR	NR	NR	NR	NR	NR	48%	
Hoffmann et al. (2015)	Germany	2001–2012	PVR:21	9 (43%)	65	NR	4 (19%)	NR	12 (57%)	12 (57%)	8 (38%)	9 (43%)	10 (48%)	20%	
			No VR:39	28 (72%)	68	NR	5 (13%)	NR	23 (59%)	17 (44%)	14 (36%)	4 (10%)	20 (51%)	18%	
Peng et al. (2016)	China	2005–2012	HAR:26	18 (69%)	59	327 ± 146	NR	15 (58%)	22 (85%)	5 (19%)	16 (62%)	NR	NR	31%	
			No VR:35	20 (57%)	63	400 ± 209	NR	15 (43%)	28 (80%)	3 (9%)	19 (54%)	NR	NR	39%	
Schimizzi et al. (2018)	United States	1998–2015	PVR:19	10 (53%)	62	NR	NR	13 (68%)	14 (74%)	3 (16%)	NR	NR	9 (60%)	NR	
			HAR:12	6 (50%)	52	NR	NR	6 (50%)	8 (67%)	0 (0%)	NR	NR	4 (33%)	NR	
			No VR:170	69 (41%)	66	NR	NR	114 (67%)	119 (70%)	7 (4%)	NR	NR	22 (15%)	NR	
Hemming et al. (2011)	United States	1999–2010	PVR:42	NR	NR	NR	NR	NR	NR	NR	NR	NR	NR	NR	
			No VR:53	NR	NR	NR	NR	NR	NR	NR	NR	NR	NR	NR	
Tamoto et al. (2014)	Japan	2005–2009	PVR:36	25 (69%)	68.5	1,902 ± 1,287	0 (0%)	21 (58%)	28 (78%)	2 (6%)	12 (33%)	11 (31%)	20 (56%)	59%	
			No VR:13	10 (77%)	68	1,980 ± 867	2 (15%)	10 (77%)	12 (92%)	2 (15%)	5 (38%)	0 (0%)	5 (38%)	51%	
Higuchi et al. (2019)	Japan	2000–2016	PVR:56	38 (68%)	69.5	NR	3 (5%)	NR	35 (63%)	NR	NR	NR	NR	NR	
			HAR:19	13 (68%)	67.0	NR	3 (16%)	NR	12 (63%)	NR	NR	NR	NR	NR	
			No VR:174	126 (72%)	70.0	NR	3 (2%)	NR	115 (66%)	NR	NR	NR	NR	NR	
Lee et al. (2010)	Korea	2001–2008	PVR:38	NR	NR	NR	NR	NR	NR	NR	NR	NR	NR	NR	
			HAR:5	NR	NR	NR	NR	NR	NR	NR	NR	NR	NR	NR	
			No VR:259	NR	NR	NR	NR	NR	NR	NR	NR	NR	NR	NR	
Igami et al. (2010)	Japan	2001–2008	PVR:69	NR	NR	NR	NR	NR	NR	NR	NR	NR	NR	23%	
			HAR:53	NR	NR	NR	NR	NR	NR	NR	NR	NR	NR	33%	
			No VR:176	NR	NR	NR	NR	NR	NR	NR	NR	NR	NR	51%	
She et al. (2020)	China	1989–2016	PVR:17
HAR:5	14 (64%)	57.0	2,875 ± 1,875	1 (5%)	11 (50%)	10 (45%)	NR	NR	9 (41%)	15 (68%)	15%	
			No VR:68	49 (72%)	67.5	1,465 ± 4,925	11 (16%)	41 (60%)	38 (56%)	NR	NR	19 (28%)	34 (51%)	27%	
Kondo et al. (2004)	Japan	1999–2002	PVR:6	NR	NR	NR	NR	NR	NR	NR	NR	NR	NR	NR	
			HAR:8	NR	NR	NR	NR	NR	NR	NR	NR	NR	NR	NR	
			No VR:26	NR	NR	NR	NR	NR	NR	NR	NR	NR	NR	NR	
De Jong et al. (2012)	United States	1984–2010	PVR:51	29 (57%)	66	NR	9 (18%)	NR	34 (67%)	NR	NR	25 (49%)	NR	NR	
			No VR:173	100 (58%)	66	NR	26 (15%)	NR	115 (66%)	NR	NR	57 (33%)	NR	NR	
Miyazaki et al. (2007)	Japan	1981–2004	PVR:34	18 (53%)	64	1,975 ± 1,474	3 (9%)	13 (38%)	NR	NR	17 (50%)	29 (85%)	NR	25%	
			HAR:9	7 (78%)	59	1,726 ± 1,253	3 (33%)	7 (78%)	NR	NR	7 (78%)	9 (100%)	NR	0%	
			No VR:118	77 (65%)	65	1,523 ± 1,147	5 (4%)	42 (36%)	NR	NR	53 (45%)	102 (86%)	NR	41%	
Muñoz et al. (2002)	United States	1990–2001	PVR:10	7 (70%)	61	NR	NR	NR	NR	NR	3 (30%)	NR	NR	22%	
			No VR:18	5 (28%)	66	NR	NR	NR	NR	NR	7 (39%)	NR	NR	38%	
Klempnauer et al. (1997)	Germany	1971–1995	PVR:40	NR	NR	NR	NR	NR	30 (73%)	NR	NR	NR	NR	32%	
			HAR:1	NR	NR	NR	NR	NR	NR	NR	NR	NR	NR	NR	
			No VR:77	NR	NR	NR	NR	NR	55 (71%)	NR	NR	NR	NR	32%	
Matsuyama et al. (2016)	Japan	1992–2014	PVR:54	39 (72%)	70	1,981 ± 1,926	2 (4%)	38 (70%)	43 (80%)	4 (7%)	27 (50%)	27 (50%)	NR	51%	
			HAR:44	27 (61%)	69	2,212 ± 2,192	4 (9%)	36 (82%)	35 (80%)	5 (11%)	26 (59%)	35 (80%)	NR
NR	51%
22%	
			No VR:74	55 (74%)	69	1,929 ± 1,387	3 (4%)	61 (82%)	55 (74%)	6 (8%)	25 (34%)	21 (28%)	NR	46%	
Yu et al. (2017)	China	2006–2014	PVR:10
HAR:9	NR
NR	55.40	NR	NR	16 (84%)	NR	0 (0%)	NR	7 (37%)	NR	0%	
			No VR:76	43 (57%)	61.03	NR	NR	45 (59%)	NR	4 (5%)	NR	12 (16%)	NR	12%	
Mizuno et al. (2020)	Japan	2001–2018	PVR:157
HAR:146	49 (31%)
NR	67
67	1,498 ± 1,805
1,491 ± 1,146	3 (2%)
2 (1%)	145 (48%)
NR	108 (69%)
93 (64%)	54 (34%)
49 (34%)	100 (64%)
91 (62%)	NR
NR	NR
NR	25%
30%	
			No VR:484	162 (33%)	69	1,078 ± 891	1 (0%)	242 (50%)	410 (85%)	102 (21%)	179 (37%)	NR	NR	50%	
Song et al. (2009)	Korea	1989–2005	PVR:51	NR	NR	NR	5 (10%)	24 (47%)	NR	NR	19 (37%)	NR	NR	19%	
			No VR:208	NR	NR	NR	6 (3%)	82 (39%)	NR	NR	48 (23%)	NR	NR	27%	
Note:

VR, vascular resection; PVR, portal vein resection; HAR, hepatic artery resection; NR, not retrievable; UICC, Union for International Cancer Control Unites.

90-day mortality

Eleven studies provided data on 90-day mortality (Higuchi et al., 2019; She et al., 2020; Yu et al., 2017; Miyazaki et al., 2007; De Jong et al., 2012; Dumitraşcu et al., 2017; Hoffmann et al., 2015; Matsuyama et al., 2016; Song et al., 2009; Tamoto et al., 2014; Wang et al., 2015). The meta-analysis indicated that VR could increase postoperative mortality (OR: 1.66, 95% CI [1.11–2.48], P = 0.01) (Fig. 2A). A significant difference also existed between the PHC patients with and without PVR, and the pooled OR (95% CI) was 1.61 (1.02, 2.54), with P = 0.04 (Fig. 2B). For patients with and without HAR, the pooled result showed significantly higher mortality among patients who underwent HAR (OR: 4.20, 95% CI [1.88–9.39], P = 0.0005) (Fig. 2C).

Figure 2 Meta-analysis of studies on 90-day mortality.

(A) 90-day mortality rate in patients with and without VR; (B) 90-day mortality rate in patients with and without PVR; (C) 90-day mortality rate in patients with and without HAR.

Overall morbidity

Eleven studies containing 2,189 patients provided data on overall morbidity (Mizuno et al., 2020; Schimizzi et al., 2018; She et al., 2020; Yu et al., 2017; Peng et al., 2016; Miyazaki et al., 2007; Ebata et al., 2003; Matsuyama et al., 2016; Song et al., 2009; Tamoto et al., 2014; Wang et al., 2015). The meta-analysis indicated no difference in morbidity between the patients with and without VR (OR: 1.04, 95% CI [0.86–1.26], P = 0.68) (Fig. 3A). A similar result was also found when comparing overall morbidity between patients with and without PVR (OR: 1.03, 95% CI [0.74–1.42], P = 0.88) (Fig. 3B). Furthermore, the meta-analysis indicated that HAR did not increase postoperative morbidity (OR: 1.32, 95% CI [0.83–2.11], P = 0.24) (Fig. 3C).

Figure 3 Meta-analysis of studies on overall morbidity.

(A) Overall morbidity rate in patients with and without VR; (B) overall morbidity rate in patients with and without PVR; (C) overall morbidity rate in patients with and without HAR.

Posthepatectomy liver insufficiency

To further explore the impact of VR on PHI, we analyzed this complication alone. Ten studies provided data on PHI (Mizuno et al., 2020; Schimizzi et al., 2018; Yu et al., 2017; Peng et al., 2016; Dumitraşcu et al., 2017; Ebata et al., 2003; Hoffmann et al., 2015; Matsuyama et al., 2016; Tamoto et al., 2014; Wang et al., 2015). The meta-analysis indicated a significantly higher PHI rate among patients with VR (OR: 1.77, 95% CI [1.37–2.28], P < 0.00001) (Fig. 4A). A similar result was obtained when comparing the PHI rate between patients with and without PVR (OR: 1.60, 95% CI [1.19–2.16], P = 0.002) (Fig. 4B). For patients with and without HAR, the pooled result showed a significantly higher PHI rate among patients who underwent HAR (OR: 1.77, 95% CI [1.23–2.54], P = 0.002) (Fig. 4C).

Figure 4 Meta-analysis of studies on posthepatectomy liver insufficiency (PHI).

(A) PHI rate in patients with and without VR; (B) PHI rate in patients with and without PVR; (C) PHI rate in patients with and without HAR.

R0 margin status

Twelve studies containing 2,294 patients reported the difference in the R0 margin status (Higuchi et al., 2019; Mizuno et al., 2020; Schimizzi et al., 2018; She et al., 2020; Peng et al., 2016; De Jong et al., 2012; Dumitraşcu et al., 2017; Ebata et al., 2003; Hoffmann et al., 2015; Klempnauer et al., 1997; Matsuyama et al., 2016; Tamoto et al., 2014). The meta-analysis indicated no difference in the R0 resection rate between patients with and without VR (OR: 0.71, 95% CI [0.50–1.01], P = 0.06) (Fig. 5A). The analysis between patients with and without PVR showed no statistically significant difference (OR: 0.70, 95% CI [0.47–1.03], P = 0.07) (Fig. 5B). For patients with and without HAR, the meta-analysis demonstrated a similar outcome (OR: 0.77, 95% CI [0.37–1.61], P = 0.49) (Fig. 5C).

Figure 5 Meta-analysis of studies on R0 margin status.

(A) R0 resection rate in patients with and without VR; (B) R0 resection rate in patients with and without PVR; (C) R0 resection rate in patients with and without HAR.

Long-term survival

Eighteen studies provided data on 1-, 3-, and 5-year OS and DFS (Mizuno et al., 2020; Schimizzi et al., 2018; She et al., 2020; Yu et al., 2017; Peng et al., 2016; Miyazaki et al., 2007; Nagino et al., 2010; Dumitraşcu et al., 2017; Ebata et al., 2003; Hoffmann et al., 2015; Igami et al., 2010; Klempnauer et al., 1997; Kondo et al., 2004; Matsuyama et al., 2016; Muñoz et al., 2002; Song et al., 2009; Tamoto et al., 2014; Wang et al., 2015). The pooled results are shown in Table 2. The pooled analysis showed that patients with VR had worse long-term survival. The meta-analysis showed that the 3- and 5-year OS rates were significantly lower in patients with VR than in those without VR (P < 0.00001), while the 1-year OS was not statistically significant different (OR: 0.94, 95% CI [0.54–1.64], P = 0.83). In addition, compared with those without PVR, patients with PVR had worse long-term survival (1-year OS: OR: 0.77, 95% CI [0.49–1.20], P = 0.25; 3-year OS: OR: 0.45, 95% CI [0.36–0.57], P < 0.00001; 5-year OS: OR: 0.52, 95% CI [0.35–0.76], P = 0.0008). For patients with and without HAR, the pooled result showed significantly worse long-term survival among patients who underwent HAR (1-year OS: OR: 0.64, 95% CI [0.11–3.69], P = 0.62; 3-year OS: OR: 0.55, 95% CI [0.41–0.74], P < 0.0001; 5-year OS: OR: 0.43, 95% CI [0.32–0.57], P < 0.00001). Meanwhile, there was no difference in the 1-, 3-, or 5-year DFS between patients with and without VR (OR: 1.54, 95% CI [0.92–2.57], P = 0.10; OR: 1.00, 95% CI [0.59–1.71], P = 0.99; OR: 0.99, 95% CI [0.42–2.35], P = 0.98). Furthermore, eight studies provided data on the HR for OS (Mizuno et al., 2020; Schimizzi et al., 2018; Nagino et al., 2010; De Jong et al., 2012; Igami et al., 2010; Kondo et al., 2004; Muñoz et al., 2002; Tamoto et al., 2014). The pooled analysis indicated that VR was relevant to a shorter OS (HR: 1.44, 95% CI [1.25–1.67], P < 0.001) (Fig. 6A).

Figure 6 Funnel plots of main results in patients with and without VR.

(A) Overall survival; (B) intraoperative blood loss; (C) proportion of III, IV stage according to UICC staging systems; (D) vascular invasion confirmed by histology.

Table 2 Meta-analysis results of pooled survival in all included studies.

	Group	I2	Pooled OR	95% CI	P value	
1-year OS	VR	55%	0.94	[0.54–1.64]	0.83	
PVR	48%	0.77	[0.49–1.20]	0.25	
HAR	78%	0.64	[0.11–3.69]	0.62	
3-year OS	VR	35%	0.56	[0.46–0.68]	<0.00001	
PVR	21%	0.45	[0.36–0.57]	<0.00001	
HAR	42%	0.55	[0.41–0.74]	<0.0001	
5-year OS	VR	27%	0.48	[0.40–0.58]	<0.00001	
PVR	54%	0.52	[0.35–0.76]	0.0008	
HAR	0%	0.43	[0.32–0.57]	<0.00001	
1-year DFS	VR	3%	1.54	[0.92–2.57]	0.10	
3-year DFS	VR	0%	1.00	[0.59–1.71]	0.99	
5-year DFS	VR	0%	0.99	[0.42–2.35]	0.98	
Note:

OR, odds ratio; CI, confidence interval; VR, vascular resection; PVR, portal vein resection; HAR, hepatic artery resection; OS, overall survival; DFS, disease free survival.

Intraoperative blood loss

Eight included studies provided data on intraoperative blood loss (Mizuno et al., 2020; She et al., 2020; Peng et al., 2016; Miyazaki et al., 2007; Dumitraşcu et al., 2017; Matsuyama et al., 2016; Tamoto et al., 2014; Wang et al., 2015), and the mean volume of blood loss was significantly greater when VR was performed (MD = 433.66, 95% CI [91.69–775.63], P = 0.01) (Fig. 6B).

UICC staging

Five studies provided data on UICC staging (Schimizzi et al., 2018; She et al., 2020; Ebata et al., 2003; Hoffmann et al., 2015; Tamoto et al., 2014). The proportion of patients diagnosed at UICC stage T3-T4 ranged from 48% to 100% and from 15% to 51% in patients with and without VR, respectively. The meta-analysis indicated a higher UICC staging among patients with VR (OR: 4.72, 95% CI [1.05–21.12], P = 0.04) (Fig. 6C).

Vascular invasion

Vascular invasion was reported in eight studies (She et al., 2020; Yu et al., 2017; Miyazaki et al., 2007; De Jong et al., 2012; Ebata et al., 2003; Hoffmann et al., 2015; Matsuyama et al., 2016; Tamoto et al., 2014), and the positive invasion rate ranged from 31% to 88% and from 0% to 86% in patients with and without VR, respectively. The mean vascular invasion rate was 39% in patients without VR, 85% in patients with PVR, and 49% in patients without HAR. Patients who underwent VR had a higher vascular invasion rate (OR: 2.31, 95% CI [1.70–3.13], P < 0.00001) (Fig. 6D).

Lymph node metastasis

Lymph node metastasis was reported in ten of the included studies (Yu et al., 2017; Peng et al., 2016; Miyazaki et al., 2007; Dumitraşcu et al., 2017; Ebata et al., 2003; Hoffmann et al., 2015; Matsuyama et al., 2016; Muñoz et al., 2002; Song et al., 2009; Tamoto et al., 2014). The mean lymph node metastasis rates in patients with and without VR were 55.5% and 35.8%, respectively. The mean lymph node metastasis rates in patients with PVR and HAR were 52.1% and 62.2%, respectively. As shown in Figs. 7A–7C, the meta-analyses revealed that patients with VR, either PVR or HAR, had a higher lymph node metastasis rate than those without VR (OR: 2.20, 95% CI [1.80–2.69], P < 0.00001; OR: 2.07, 95% CI [1.64–2.61], P < 0.00001; OR: 2.68, 95% CI [1.95–3.68], P < 0.00001, respectively).

Figure 7 Meta-analysis of studies on lymph node metastasis.

(A) Lymph node metastasis rate in patients with and without VR; (B) lymph node metastasis rate in patients with and without PVR; (C) lymph node metastasis rate in patients with and without HAR.

Discussion

PHC is a rare malignancy that accounts for less than 2% of total human malignancies (Cai et al., 2011). The tumor often invades the bile duct through the wall and extends to the periductal tissues and adjacent structures (Hayashi et al., 1994). Given the anatomical location and aggressive biological characteristics of PHC, most PHC patients are in advanced stages when examined. In fact, despite the use of various imaging tests to assess the tumor status, 40–50% of PHC patients are found to have unresectable tumors during the operation (Parikh, Abdalla & Vauthey, 2005; Ruys et al., 2011). Among them, involvement of the main portal vein, bilateral portal vein and/or hepatic artery branches are important reasons for the unresectability of tumors (Parikh, Abdalla & Vauthey, 2005).

Surgical resection for PHC is highly technically demanding and could be challenging for hepatobiliary surgeons (Nagino, 2012). Due to the changes in surgical philosophy and other aspects, radical surgery for PHC has also undergone great changes. Currently, curative surgery for PHC includes major hepatectomy, bile duct excision, locoregional lymph node dissection, and combined caudate lobectomy (Lee et al., 2010; Nagino et al., 2013; Mansour et al., 2015). Due to local anatomical considerations, vascular invasion is not uncommon in PHC. According to the included studies, the rate of vascular invasion confirmed by histology ranges from 20% to 87% (She et al., 2020; Yu et al., 2017; Miyazaki et al., 2007; De Jong et al., 2012; Ebata et al., 2003; Hoffmann et al., 2015; Matsuyama et al., 2016; Tamoto et al., 2014). Furthermore, when the vessel can be reconstructed after resection, vascular invasion is no longer an absolute contraindication for PHC surgery. However, while VR (including PVR and HAR) has been performed at many clinical centers, their effect in patients with PHC remains controversial, and previous comparative studies have reported inconsistent results (Miyazaki et al., 2007; De Jong et al., 2012; Lee et al., 2010; Nagino et al., 2013; Neuhaus et al., 1999; Neuhaus et al., 2012; Hirano et al., 2010).

Due to the complexity of biliary and hepatic resection, the postoperative morbidity rate for PHC is significant, ranging from 36% to 81% (Mizuno et al., 2020; Schimizzi et al., 2018; She et al., 2020; Yu et al., 2017; Peng et al., 2016; Miyazaki et al., 2007; Ebata et al., 2003; Song et al., 2009; Tamoto et al., 2014; Wang et al., 2015). This meta-analysis showed that neither PVR nor HAR increased the incidence of postoperative complications (all P > 0.05). PHI seriously affects the patient’s recovery and prognosis. The meta-analysis indicated that patients with PVR had a significantly higher incidence of PHI, and a similar result could be found when comparing patients with and without HAR. The reasons for these findings are that PVR and/or HAR may prolong the period of liver ischemia during vascular reconstruction, which may aggravate ischemic damage to the remnant liver (Zhang et al., 2015). To reduce the incidence of PHI, preoperative portal vein embolization (PVE), which was first proposed by Kinoshita et al. (1986) and Makuuchi et al. (1990) in the 1980s, has been widely performed in many centers before surgery for PHC.

Whether PVR increases postoperative mortality remains controversial. The portal vein bifurcation lies directly posterior to the hepatic duct confluence and therefore frequently shows tumor involvement. To achieve R0 resection, curative surgery might therefore require concomitant resection of the portal vein bifurcation. Most studies have indicated that patients with PVR have a higher mortality rate than those without PVR, ranging from 0% to 19% and from 0% to 16%, respectively (Higuchi et al., 2019; Mizuno et al., 2020; Miyazaki et al., 2007; De Jong et al., 2012; Dumitraşcu et al., 2017; Hoffmann et al., 2015; Matsuyama et al., 2016; Song et al., 2009; Wang et al., 2015), but other studies have shown inconsistent results. In 2014, Tamoto et al. (2014) reported 0% mortality in patients with PVR and 15% mortality in patients without PVR, which was similar to She’s study (She et al., 2020). This meta-analysis showed that PVR might increase mortality. However, the mean mortality rate was 4.0% in patients without PVR and 6.2% in patients with PVR. These results showed that although PVR increased mortality, it was to an acceptable level. Compared to PVR, the effect of HAR on mortality was similar. All five included studies showed that patients with HAR had a higher mortality rate than those without HAR. The meta-analysis showed that HAR greatly increased mortality (P = 0.0005). The mean mortality rate was 1.7% in patients without HAR and 5.4% in patients with HAR. Consequently, it seems that HAR is more likely to significantly increase mortality.

The resection margin is a vital prognostic factor for PHC surgery. In most surgical series that have included patients treated with hepatectomy combined with extrahepatic biliary resection, an R0 margin was obtained in 55–90% of patients (Higuchi et al., 2019; Mizuno et al., 2020; Schimizzi et al., 2018; She et al., 2020; Peng et al., 2016; De Jong et al., 2012; Dumitraşcu et al., 2017; Ebata et al., 2003; Hoffmann et al., 2015; Klempnauer et al., 1997; Matsuyama et al., 2016; Tamoto et al., 2014). Although R1 resection has shown some benefit to survival when compared to nonoperative treatment, R0 margins should be achieved as far as possible (Baton et al., 2007; Hidalgo et al., 2008; Lee et al., 2012). This meta-analysis showed no difference in the R0 resection rate between patients with and without PVR, and a similar result could be found when comparing patients with and without HAR. The mean R0 resection rates were 76%, 69% and 70% in patients without VR, with HAR and with PVR, respectively. Although patients with VR had disease of a more advanced stage, the validity of VR in terms of obtaining a better surgical margin still should be considered in such patients. Combined with previous studies, we seem to be able to conclude that VR (including PVR and HAR) can achieve a higher R0 resection rate because these patients can only achieve R1 resection or even R2 resection if VR is not performed. Of course, this conclusion needs to be further verified.

The results of the survival analysis showed that patients with PVR had poorer OS than those without VR, although the 1-year OS was not statistically significant different. These results seem to imply that the surgical oncologic outcome of patients with PVR is worse than that of patients without PVR. However, subsequent analysis found that patients with PVR had more advanced disease and higher positive lymph node metastasis, both of which are adverse prognostic factors (Lurje et al., 2019). Furthermore, some studies have shown that patients with PVR have a significant survival advantage over unresectable patients (She et al., 2020; Nagino et al., 2010; Ebata et al., 2003). Considering that PVR did not increase the postoperative morbidity rate and slightly increased the mortality rate, it seems that PVR is acceptable for selected patients. At present, there is no uniform conclusion on the selection of suitable patients for PVR. The invasion of both portal branches strongly contraindicates hepatic resection and this has been sustained by different guidelines (Miyazaki et al., 2015; Rizvi et al., 2018; Banales et al., 2020). In addition, patients with distant metastatic disease or involvement of aortocaval or truncal nodes are unlikely to benefit from resection (Groot Koerkamp et al., 2014). Therefore, we suggest that PVR could be performed in PHC patients with a preoperative or intraoperative finding of unilateral portal vein invasion and without a distant metastatic event. Besides, considering the increased incidence of PHI associated with PVR, surgical decision should be made cautiously according to the physical condition of patients. However, the meta-analysis showed that HAR did not increase postoperative morbidity and achieved an acceptable R0 resection rate but significantly increased postoperative mortality. Meanwhile, for long-term survival, the 1-, 3-, and 5-year OS rates in patients with HAR were 59.57%, 43.90% and 27.81%, respectively, and 64.71%, 54.12% and 46.75% in patients without HAR, respectively. These results showed that HAR has not been demonstrated to benefit PHC patients in terms of safety and long-term survival.

High heterogeneity was found in the analysis of several covariates, especially R0 margin status (I2 = 70, P = 0.009), intraoperative blood loss (I2 = 89%, P < 0.00001) and UICC staging (I2 = 85%, P < 0.0001). For R0 margin status, through a “leave-one-out” sensitivity analysis, we found that one study (Mizuno et al., 2020) may have contributed to the heterogeneity. In Mizuno’s study, patients without VR had earlier tumor statuses, with a significantly lower proportion of T4 stage patients than those with VR (either PVR or HAR), at 25% versus 85%, respectively. Therefore, the R0 resection rate in patients without VR was markedly higher than in those with VR, either PVR or HAR (84.7%, 68.8% and 63.7%, respectively). Moreover, the sample size of the study was extremely large, and therefore the effect on heterogeneity was large. After removing Mizuno’s study, similar results were obtained that neither PVR nor HAR improved the R0 resection rate. In addition, for the high heterogeneity found in the analysis of intraoperative blood loss, the possible reasons were as follows: (1) the year of publication of the included studies ranged from 1997 to 2020, and advances in surgical techniques across this relatively long period could lead to large differences in intraoperative parameters, such as intrahepatic blood loss; (2) surgical experience varies among clinical centers, and intraoperative blood loss thus varies among different centers; and (3) although all PHC patients underwent hepatectomy, the extent of liver resection varied depending on the location of the tumor, thus resulting in a difference in intraoperative blood loss. Likewise, for the obvious heterogeneity found in the analysis of UICC staging, after checking the details, we found that two studies (Schimizzi et al., 2018; Ebata et al., 2003) may have contributed to the heterogeneity. In these studies, a much higher proportion of patients with VR were diagnosed at UICC stage T3-T4.

This review has several limitations that should be mentioned. First, there were no randomized trials on this topic, and all eligible studies were observational studies. Second, a large number of studies were excluded due to either inadequate data or the lack of an effective comparison group. Third, data were missing in a few of the included studies, and the statistical power was relatively low. Last, the retrospective study design has inherent limitations, and inherent information bias in the original studies can always cause problems.

Conclusions

In conclusion, PHC is an uncommon and aggressive disease with a poor long-term prognosis. PVR is relatively safe and might confer benefits to certain patients with advanced PHC in terms of long-term survival. HAR is related to increased mortality and has not been demonstrated to benefit long-term survival, which should be considered before performing this procedure. Data from randomized controlled trials are required to further prove the findings in this study.

Supplemental Information

Supplemental Information 1 Meta-analysis of studies on 1-year OS

(A) 1-year OS in patients with and without VR; (B) 1-year OS in patients with and without PVR; (C) 1-year OS in patients with and without HAR.

Click here for additional data file.

Supplemental Information 2 Meta-analysis of studies on 3-year OS

(A) 3-year OS in patients with and without VR; (B) 3-year OS in patients with and without PVR; (C) 3-year OS in patients with and without HAR.

Click here for additional data file.

Supplemental Information 3 Meta-analysis of studies on 5-year OS

(A) 5-year OS in patients with and without VR; (B) 5-year OS in patients with and without PVR; (C) 5-year OS in patients with and without HAR.

Click here for additional data file.

Supplemental Information 4 Meta-analysis of studies on DFS in patients with and without VR

(A) 1-year DFS; (B) 3-year DFS; (C) 5-year DFS.

Click here for additional data file.

Supplemental Information 5 Forest plots of main results in patients with and without VR

(A) 90-day mortality rate; (B) overall mortality rate; (C) PHI rate; (D) R0 resection rate; (E) 1-year OS; (F) 3-year OS; (G) 5-year OS; (H) 1-year DFS; (I) 3-year DFS; (J) 5-year DFS; (K) proportion of Ⅲ、Ⅳ stage according to UICC staging systems; (L) positive vascular invasion rate; (M) mean lymph node metastasis rate; (N) intraoperative blood loss .

Click here for additional data file.

Supplemental Information 6 Forest plots of main results in patients with and without PVR

(A) 90-day mortality rate; (B) overall mortality rate; (C) PHI rate; (D) R0 resection rate; (E) 1-year OS; (F) 3-year OS; (G) 5-year OS; (H) mean lymph node metastasis rate.

Click here for additional data file.

Supplemental Information 7 Forest plots of main results in patients with and without HAR

(A) 90-day mortality rate; (B) overall mortality rate; (C) PHI rate; (D) R0 resection rate; (E) 1-year OS; (F) 3-year OS; (G) 5-year OS; (H) mean lymph node metastasis rate.

Click here for additional data file.

Supplemental Information 8 Raw information of the studies included in the current meta-analysis

Click here for additional data file.

Supplemental Information 9 Quality of the included studies assessed by the Newcastle-Ottawa scale (NOS)

Click here for additional data file.

Supplemental Information 10 Search strategy

Click here for additional data file.

Supplemental Information 11 PRISMA checklist

Click here for additional data file.

Supplemental Information 12 Rationale and contribution

Click here for additional data file.

The authors thank American Journal Experts for language editing.

Additional Information and Declarations

Competing Interests

Author Contributions

Data Availability

The authors declare that they have no competing interests.

Yong Liu conceived and designed the experiments, performed the experiments, authored or reviewed drafts of the paper, and approved the final draft.

Guangbing Li conceived and designed the experiments, performed the experiments, analyzed the data, authored or reviewed drafts of the paper, and approved the final draft.

Ziwen Lu conceived and designed the experiments, performed the experiments, analyzed the data, prepared figures and/or tables, and approved the final draft.

Tao Wang analyzed the data, prepared figures and/or tables, and approved the final draft.

Yang Yang analyzed the data, prepared figures and/or tables, and approved the final draft.

Xiaoyu Wang analyzed the data, prepared figures and/or tables, authored or reviewed drafts of the paper, and approved the final draft.

Jun Liu conceived and designed the experiments, authored or reviewed drafts of the paper, and approved the final draft.

The following information was supplied regarding data availability:

The raw measurements are available in Supplemental Files.

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
