# Peer review of "Effect of vascular resection for perihilar cholangiocarcinoma: a systematic review and meta-analysis"

_PeerJ, doi:10.7717/peerj.12184_

## Round 0.1 · original submission · Major Revisions

Please attempt to better distinguish venous and arterial resections in order to improve your paper's clinical relevance.

·

Basic reporting

- Peri-hilar cholangiocarcinoma should be used instead of hilar cholangiocarcinoma (Cholangiocarcinoma Working Group, Dig Liv Dis 2020)

- I can't find any of the following ref: previous meta-analyses Wu 2013, Chen 2014, Abbas 2013, nor the last italian guidelines (see above). wu and chen's reviws were only on PV resection, whereas Abbas' was on vascular resection generally

- Manuscript would (really) benefit of a review by a native English speaker

- Fig 4B is cited after FIg 4C, Fig 4A is not cited in the results section

- Fig 4: legends are not clear, the authors should state in which direction vasc resection or no vascular resection favors (i.e. I have the impression that no vasc resection favors stage 3/4 UICC, but when I read your text it is quite the opposite) ; maybe the scale of fig 4C should be changed so we can actually see all the plots?

- for more clarity, an axis "favors XXXX" should be added to each funnel plot so the reader easily understand what vasc resec favors

Experimental design

- Introduction/research question/discussion : I think that portal vein resection and hepatic artery resection are totally different attitudes. However, the authors do not distinguish between both in the introduction. On the one hand, there is a intimacy in the hepatic pedicle between hilar confluence and PV, does routine PV resection improves oncological results without significant increase in morbidity? on the other hand, HA resection is a rare event and is performed mostly in highly selected patient, sometimes with anatomical variations.

- in table 1, it would be useful to distinguish PV resection/HA resection/PV+HA resection in the numbers of patients assessed

- methods and ethics are OK

Validity of the findings

- The rationale distinguishing PV from HA should be clearer both in introduction, results and discussion

Additional comments

There are not many meta-analyses on that specific topic, the last one being Abbas' in 2014. The present meta-analysis has almost 2,000 patients more and newer studies. However, from my surgical perspective, (i) it does not bring new data to the pre-existing literature, (ii) it would be very important to distinguish between PV and HA resection. As for now, I think that most surgeons agree that a PV resection should be made in specialized centers when deemed necessary because of suspicion of PV encasement, and that HA resection should be either not recommended or reserved to very selected cases. Introduction, aims, methods, results and discussion should be made according to PV or HA resection, but not "vascular resection".

Minor remarks
- line 50: resection surgery and transplantation
- line 178: capital letter is missing
- please review English

Reviewer 2 ·

Basic reporting

Authors could provide more clarity on the primary and secondary outcomes of the study. I would suggest an additional paragraph in the Methods section of the manuscript detailing the primary and secondary outcomes that the authors intended to study in this meta-analysis, together with definitions for these outcomes. For instance, it is not precisely apparent what the authors mean by “total morbidity” or “hepatic insufficiency”.

There were clerical errors in the paper:
1. Line 129: Figure 2B shows a total of 2189 participants in the study, but the authors report 11 studies with 1182 patients in total. Please clarify.

Certain phrases were cumbersome and could use some editing. Kindly make edits to the following:
1. Line 59: “efficacy and security of vascular resection” – the authors might have meant safety instead of security. If I have misinterpreted what the authors meant to convey, please make edits correspondingly
2. Line 62: “efficacy and security of vascular resection” – see above
3. Line 148: “who did and did not treated with” – please paraphrase. This is grammatically incorrect. Something along the lines of “between patients who were treated with and not treated with” might be acceptable.
4. Line 151: “who did and did not treated” – see above
5. Line 159: “who did and did not treated” – see above
6. Line 160: “who treated with hepatic” – please paraphrase.
7. Line 181: “just simple bile duct section initially” – please paraphrase.

Experimental design

Line 112: Article selection process was not immediately clear. In figure 1, before the authors reviewed the abstracts to decide if they were suitable for inclusion, 642 records were deemed irrelevant and excluded. This represents more than half of the initial number of records that were identified after duplicates were removed. Kindly elaborate on why these records were excluded and how they were deemed to be irrelevant. Article selection is of paramount importance and when improperly performed, would render the conclusions derived naught. I would suggest to the authors that they use the PRISMA Flow diagram, accessible at http://prisma-statement.org/prismastatement/flowdiagram.aspx, to report their article selection process. I would suggest to the authors to elaborate further on their article selection process in the manuscript as well.

Line 94: Could the authors elaborate on the methods used to approximate the Hazard Ratios when they were not immediately available in the article?

Line 105: I did not note the use of any inverse-variance methods for meta-analysis of dichotomous variables in the manuscript. Could the authors clarify when this method was employed?

Line 106: Suggest to the authors include Mean Difference as an effect measure in their statement for completeness. Something along the lines of “… forest plots and presented as Odds Ratios and Mean Differences and 95%CIs” would be acceptable.

Lines 240 to 245: The authors highlighted important limitations in their discussion. Could they elaborate on some of the processes that they undertook to overcome these limitations when conducting their study?

Validity of the findings

Line 144: The meta-analysis of Intraoperative blood loss is quite heterogenous and has an I2 of 89%. Could the authors kindly investigate further and discuss why this might be the case.

Line 160 and 161: “hepatic artery resection and reconstruction showed lower survival rates”. This statement, at face value, is not supported by the findings laid out by the authors in Figure 3E, which shows statistically insignificant differences in the Odds of survival at 1- and 3-year intervals. I would suggest to the authors to qualify or retract this statement.

Line 231: The meta-analysis of R0 resection margin is quite heterogenous and has an I2 of 70%. Could the authors kindly investigate further and discuss why this might be the case.

---

## Round 0.2 · Minor Revisions

Just a little bit more to do - please respond to these remaining comments from R2

Reviewer 2 ·

Basic reporting

Appreciate the response
Clarity of text has significantly improved

Experimental design

Queries adequately addressed and changes reflected in the manuscript

Validity of the findings

1. Analysis of high UICC stage, Vascular Invasion and Lymph Node metastasis seem to suggest that more advanced disease could be a confounder for poorer clinical outcomes. The authors' own analyses showed that these factors were more commonly present in patients undergoing vascular resection, yet these factors could also independently lead to poorer oncological outcomes, as evidenced by other authors in the past (eg Lurje et al. 10.1016/j.ejso.2019.04.019). Authors themselves also appear to be cognizant of this possibility - as evidenced by authors' arguments within lines 297 to 300 of the manuscript. I would suggest to the authors that it may be possible to implement further statistical methods to see if poorer clinical outcomes were indeed attributable to the procedural risks incurred by undergoing vascular resection, such was by performing subgroup analyses or meta-regression. However, I am also appreciative of the fact that there may not have been enough data for this to be possible, especially in the context of hepatic artery resection, where there are already limited number of studies for each meta-analysis. Suggest the authors to further look into this.

2. Authors suggested in line 304 that PVR may be acceptable for selected patients - suggest to the author to elaborate upon which patients may belong to this category.

Additional comments

-

---

## Round 0.3 · accepted · Accept

Acceptable for publication.